



# Brief Communication: Heterogenous thinning and subglacial lake activity on Thwaites Glacier, West Antarctica

Andrew O. Hoffman[1], Knut Christianson[1], Daniel Shapero[2], Benjamin E. Smith[2], Ian Joughin[2]

[1]Department of Earth and Space Sciences, University of Washington, Seattle, 98115, United States of America
[2]Applied Physics Laboratory, University of Washington, 98115, United States of America

*Correspondence to*: Andrew O. Hoffman (hoffmaao@uw.edu)

**Abstract.** A system of subglacial lakes drained on Thwaites Glacier from 2012-2014. To improve coverage for subsequent drainage events, we extended the elevation and ice velocity time series on Thwaites Glacier through austral winter 2019. These new observations document a second drainage cycle and identified two new lake systems located in the western tributaries of Thwaites and Haynes Glaciers. *In situ* and satellite velocity observations show temporary < 3% speed fluctuations associated with lake drainages. In agreement with previous studies, these observations suggest that active subglacial hydrology has little influence on Thwaites Glacier thinning and retreat on decadal to centennial timescales.

## 1 Introduction

Although subglacial lakes beneath the Antarctic Ice Sheet were first discovered more than 50 years ago (Robin et al., 1969; Oswald and Robin, 1973), they remain one of the most enigmatic components of the subglacial hydrology system. Initially identified in ice-penetrating radar data as flat, bright specular reflectors (Oswald and Robin, 1973; Carter et al., 2007) subglacial lakes were thought to be relatively steady-state features of the basal hydrology system with little impact on the dynamics of the overlying ice on multi-year timescales. The advent of high-precision repeat satellite observations in the 1990s, however, revealed an entirely new class of active subglacial lakes that fill and drain on annual to decadal timescales and possibly affect the flow of the overlying ice (e.g. Gray et al., 2004; Wingham et al., 2006; Fricker et al., 2007; Smith et al. 2010). Under the central trunk of Thwaites Glacier, in particular, satellite radar altimetry revealed a large (~4 km$^3$ volume discharge), connected subglacial-lake drainage event from 2012–2014 (Smith et al., 2017). Initial subglacial lake recharge estimates suggested lake drainages of this magnitude should occur every 20-80 years. To better constrain refill and discharge time, we extended the Thwaites altimetry record to include the most recent drainage events. We also expanded the spatial extent of coverage to include Haynes Glacier and the western tributary of Thwaites (Fig. 1). Here we describe the recent subglacial lake behaviour in these regions and discuss the impact of these subglacial lake systems on slip velocity at the ice-bed interface.



## 2 Methods

We used *in-situ* Global Navigation Satellite System (GNSS), satellite synthetic aperture radar (SAR), and satellite radar
altimetry data to derive velocity and elevation-change time series.

### 2.1 Ice Velocity and Vertical Displacement

We used the speed anomalies recorded by two long-term on-ice GNSS receivers, LTHW and UTHW, deployed on Thwaites
Glacier from 2009 to present to augment Eulerian velocity products derived from Sentinel-1A and Sentinel-1B synthetic
aperture radar (SAR) imagery of the Thwaites Catchment collected between 2015-2019 (Fig. 1). Processing of GNSS data
follows the workflow of Christianson et al. (2016). We first created a position time series with sub-5-cm uncertainties in all
dimensions by calculating geodetic positions every 30 seconds relative to two fixed rock sites located 200 km (Backer Island)
and 300 km (Howard Nunatak) away, using differential carrier phase positioning as implemented in the Track Software (Chen,
1998). We then constructed velocity time series from these geodetic solutions using 3-day Savitzky-Golay filtered moving
averages. Finally, we subtracted the SAR-derived Eulerian speed at each GNSS position to solve for the Lagrangian velocity
anomaly relative to the reference velocity field shown in Figure 1.

Following the methods of Joughin (2018, updated 2019), Joughin et al. (2010), and Joughin et al. (2018), we constructed a
speckle-tracked velocity time series of Thwaites Glacier from 2015 through the austral winter of 2019 using SAR data collected
by the European Union's Copernicus Sentinel-1A and Sentinel-1B satellites and processed by the European Space Agency.
We also computed the component of motion in the satellite line-of-sight (LOS) direction. The bulk of this signal is due to
relatively steady horizontal displacements, but it is also influenced by potentially more temporally variable vertical
displacements. Thus, we computed the mean LOS component for the full time series and subtracted it from each individual
estimate. Since the horizontal and surface-parallel flow components are relatively steady, the residual line-of-sight estimate
should largely be due to vertical motion, which we corrected for incidence angle effects to produce an approximate vertical
displacement rate ($V_z$). Because we only subtracted the mean, a small component of the horizontal velocity may map into the
vertical velocity due to the glacier's acceleration; however, this contribution is generally in the noise during periods with no
lake activity and small relative to the vertical signal during times of active lake change (Fig. 2). To more tightly constrain the
timing of the drainage events, we spatially interpolated the time series of Sentinel-1 derived $V_z$ to fill gaps in coverage and
integrated the result during a period of filling/draining (see vertical bars in Fig. 2) to produce estimates of net uplift and
subsidence shown in Figure 2a.

### 2.2 Elevation and Lake Volume Change

We also extended the previous time series of ESA CryoSat-2 radar altimetry data (Smith et al., 2017) through austral winter
2019, as shown in Figures 2b and 3. Elevation models were derived from the algorithm described by (Smith et al., 2017) by



fitting surfaces of elevation change to CryoSat-2 swath-processed elevation retrievals and points-of-closest-approach relative
to a reference elevation model from the first quarter of 2011. The fitting procedure minimized an objective functional that
considered data misfit, spatial gradients in the constructed reference DEM, elevation-change rate fields, temporal gradients in
elevation-change rate, and the magnitude of model bias parameters. In this scheme, three expected elevation statistics are used
to choose weight parameters that regularize the least squares fit. The elevation statistics, $E\left(\frac{\partial^2 z_0}{\partial x^2}\right)$, $E\left(\frac{\partial^3 z}{\partial x^2 \partial t}\right)$,
$E\left(\frac{\partial^2 z}{\partial t^2}\right)$ represent thresholds for spatial and temporal derivatives of the reference elevation model, $z_0$, and the time dependent
height-change field, $z$. The values chosen for this study are $E\left(\frac{\partial^2 z_0}{\partial x^2}\right) = 6.7 \times 10^{-8} \text{m}^{-2}$, $E\left(\frac{\partial^3 z}{\partial x^2 \partial t}\right) = 6 \times 10^{-9} \text{m}^{-2}\text{yr}^{-1}$ ,
$E\left(\frac{\partial^2 z}{\partial t^2}\right) = 1.0 \text{m/yr}^2$, which tighten the spatial variations in the least square's elevation-change time series compared to the
original Smith et al (2017) paper by factors of 5 and 10, respectively. These radar altimetry measurements complement
underused SAR observations of integrated vertical displacement, which we use together to understand the character of new
lake drainage activity.

## 3 Results: new observations of lake activity

A complete chronology of progressive thinning and lake activity across Thwaites Glacier from the extended CryoSat-2 time
series is shown in the video supplement (SV1). These new observations reveal that the upper Thwaites Lakes, Thw$_{170}$, and
Thw$_{142}$, drained in 2017, filling Thw$_{124}$. The SAR-derived elevation-change data show that the largest lake, Thw$_{124}$, filled by
1.9km$^3$ during the 2017 drainage, roughly balancing the volume that drained from Thw$_{142}$ (0.6km$^3$) and Thw$_{170}$ (1.4km$^3$). The
quarterly CryoSat-2 results show less clear evidence of balance (Fig 2, Fig. S2), which may be due to the degree of smoothing
used in producing the time series. From CryoSat-2 elevation-change data, between 2015 and before the 2017 drainage event,
the areas inside the Thw$_{124,}$ Thw$_{142}$, and Thw$_{170}$ outlines increase in elevation, which is strong evidence of filling (Fig. 3, Fig
S2).

The extended elevation time series (Fig. 3, Fig S2) also reveals the fill-drain cycle of two new lake systems: one in the western
shear margin of Haynes Glacier and another in the western tributary of Thwaites Glacier (Fig 1). From these combined
observations, the western Thwaites tributary lake drained by 1.1km$^3$ in 2013 and the Haynes Glacier lake system drained by
0.2km$^3$ in 2017 (Figs. S1 & S2). Complete fill-drain cycles of the Haynes Glacier lakes and the western Thwaites Glacier are
not observed in the existing altimetry record and the Haynes Glacier lakes do not discernibly refill after draining in 2017 (Fig.
S2). The western Thwaites tributary lake, however, fills significantly after draining in 2014 at a rate of 0.1km$^3$/yr.

## 4 Discussion

Cascading lake drainages have been observed under many Antarctic ice-stream systems (Wingham et al., 2006; Fricker et al.,
2007; Siegfried et al., 2014; 2016). The positions of all identified lakes beneath Thwaites Glacier, including the new lakes in



the Haynes Glacier shear margin and western tributary of Thwaites Glacier appear to be controlled primarily by the bed and
associated surface geometry (Smith et al., 2017). There are large topographic ridges at the bed with corresponding expressions
at the surface that are oriented orthogonal to flow and likely act as hydraulic baffles trapping water and sediments (Holschuh
et al., 2020), causing hundred-kPa-scale deviations in basal traction (Joughin et al., 2014; S4). The weak basal shear stress in
these till-draped basins combine with large scale catchment topography to promote variations in ice thickness and surface
slope that form large hydropotential lows (Fig. S3, Smith et al., 2017; Holschuh et al., 2020). In these hydropotential lows, the
lakes remain disconnected from their neighbours as they fill until cascading drainages driven by upstream lakes interrupt the
background fill rate in the cycle. Densely-sampled SAR vertical displacement rates from 2017 ($V_z$ in Fig. 2c) demonstrate this
process, capturing the $Thw_{170}$ drainage that initiated a combined drainage with $Thw_{142}$ into $Thw_{124}$ (Fig. 2).

The controls on lake filling are less clear. From the complete fill cycle we observe for $Thw_{170}$, the average fill rate is
~0.16km³/yr (Fig. S3). This roughly agrees with the fill rate (~0.14km³/yr) we calculate routing inferred basal meltwater
production (Joughin et al., 2009) down the Shreve hydropotential gradient (Shreve, 1972) into the lake but requires efficient
inflow of all melt water produced upstream of $Thw_{170}$ into the $Thw_{170}$ lake basin (Fig. S3). The basal water routing predicted
by the Shreve potential also route water around $Thw_{170}$ into downstream lakes $Thw_{142}$ and $Thw_{124}$, but the independent fill
rates on these flow paths (~0.44km³/yr and ~0.27km³/yr, respectively) are much larger than what we observe (Fig. S3). These
discrepant observations may reflect limitations of the static hydropotential assumption and the catchment meltwater budget,
but also suggest current bed-elevation models do not resolve small-scale (<1 km) bed topography important for routing
subglacial water into the upper most lake, $Thw_{170}$.

**4.1 Lake impact on ice flow and coupled drainage morphology**

The inland SAR and GNSS observations show a general pattern of acceleration at the LTHW and UTHW sites, consistent with
an increase in driving stress due to inland propagation of thinning caused by ungrounding and loss of ice-shelf buttressing
(Rignot et al., 2014; Joughin et al., 2014). Figure 3 shows the velocity anomaly at LTHW after subtracting the 2010 glacier
velocity (340m/yr). At LTHW, this secular trend is punctuated by two signals associated with the Thwaites Lakes drainage
events, first in September 2012 and again in May 2017 (Fig. 3). During the 2012 drainage documented by Smith et al (2017),
surface velocities initially spike by 2% over a several-day period but then decline by 3% over the following 6 months. Loss of
receiver power interrupted this record in March 2013. When the receiver began telemetering data again in 2015, the relative
change in position suggests the speed anomaly in 2014 remained below what would be expected from the 2010–2012 trend.
From Jan 2016 to May 2017, the LTHW receiver continued to accelerate at a rate of 4m/yr². As $Thw_{142}$ and $Thw_{170}$ drained in
the austral winter 2017, filling $Thw_{124}$, there was a nearly step-wise 1% increase in glacier speed at LTHW (Fig. 3). Speed
remained elevated at the LTHW site after $Thw_{124}$ stopped filling coinciding with a 2-degree shift in ice-flow direction to the
grid-north, toward $Thw_{124}$ (Fig. 3). The UTHW site also exhibits subtle velocity fluctuations with a magnitude less than 1% of



the mean velocity (105m/yr) after correcting for the spatial ice-velocity gradient. These fluctuations are small relative to the background acceleration we observe at the UTHW site, 0.75m/yr$^2$.

The velocity anomalies observed at LTHW are likely related to subglacial lake dynamics. During lake filling in 2017, the areal
extent of the Thw$_{124}$ lake increases. This increase should reduce traction in the lake boundary, both at the margins (near the LTHW site) of the lake but also inside the lake as unresolved topographic pinning points are submerged. In 2017, changes in basal traction are realized almost immediately, causing a step increase in velocity. As the lakes drain, traction is restored as ice regrounds, reducing basal slip, which we observe in the months following the 2012 lake drainage near Thw$_{124}$.

The positive GNSS acceleration observed in September 2012 (Fig. 3) suggests that Thw$_{124}$ began to drain in 2012 before the quarterly-resolved elevation fall of the lake drainage becomes distinguishable in the CryoSat-2 surface elevation time series (Fig. 3). The cause of the acceleration we observe at Thw$_{124}$ before the bulk drainage cannot be unambiguously attributed to a set of processes with these data, but the finite duration of the velocity change (~10 days) suggests that a distributed drainage system may have been established at the downstream edge of the lake preceding bulk drainage. Glaciostatic hydraulic water
routing indicates that Thw$_{124}$ would likely drain at the grid-south edge of the lake near the LTHW GNSS (Fig. S3). Enhanced lubrication outside the till-filled lake basin as the Thw$_{124}$ lake begins to drain likely increases local slip and drives the subtle change in ice-flow direction that we observe in the austral winter of 2012 before the peak drainage in 2013 when flow direction shifts back to the mean flow direction before the lake drained. Non-steady effective pressure likely also affects the basal shear stress as the drainage system initially forms, then empties and closes. Similar to lake drainages under the Siple Coast ice
streams (Siegfried et al., 2014; 2016), changes in basal slip speed due to lake activity are small (~1–2%) relative to the average sliding speed.

Regardless of the exact drainage mechanism, the locations of the lakes are governed by ice-flow response to the underlying bed topography, which promotes hydropotential basins behind seals that form as ice flows over ridges. The lake fill-drain
cycles sequester water but have little effect on the flow behaviour of the overlying ice. Once a connected drainage begins, differences in water pressure in the conduits between lakes promote efficient drainage down the hydraulic gradient. During drainage, each lake is likely in local equilibrium with the ends of the conduits that directly connect to it; however, the large hydropotential differences between lakes (~1MPa) cannot equilibrate along the entire length of the drainage path, which implies that the pressure difference over the length of the conduit is likely more important in determining whether water flows
into or out of the lakes than the small variations in the hydraulic potential at the lakes as they fill and drain. Conversely, drainage of upstream lakes may disrupt the steady-state drainage morphology by temporarily increasing the conductivity of the conduits, bringing additional water into the lower lakes, and allowing drainage from adjacent lakes that share the same downstream conduit. The processes governing changes in subglacial hydraulic connectivity are poorly understood, but likely depend on local dynamic hydropotential and the expansion or shrinkage of conduits, which we do not try to infer from the fill-



drain cycle. We only note that these processes likely contribute to the variability in lake fill-drain levels observed elsewhere over multiple fill-drain cycles (Fig S2., Siegfried et al., 2014; 2016), which cannot be explained by the evolving static hydropotential alone.

## 4.2 Implications for basin-wide change

The velocity changes (< ~3%) that we attribute to lake drainage events represent the dominant, albeit small, inland sub-annual velocity variability we measure with GNSS and SAR imagery. On decadal-timescales important for understanding ice-sheet behaviour and contribution to sea level, however, the lakes do not appear to control inland ice-flow variation. Static inversions for bed resistance before and after the 2017 lake drainage event (see Supplement for details) are not sensitive to the subtle surface velocity changes we measure with GNSS (Fig. S4), and the lakes identified by Smith et al. (2017) have no discernible

effect on ice velocity at the UTHW site. These new observations suggest that the observed speed-up at the grounding zone of the main trunk of Thwaites Glacier following the 2013 drainage (Smith et al., 2017) was associated with warming ocean conditions following anomalous Amundsen Sea wide ocean cooling from 2012-2013 (Christianson et al., 2016). These warm ocean conditions likely enhanced sub-ice-shelf melt and led to increased ungrounding and acceleration. Our observations and model experiment (see supplement) invalidate proposed geoengineering solutions that seek to drain large volumes of water

from beneath Amundsen Sea glaciers to increase basal resistance (Moore et al., 2018). These results further demonstrate that capturing the details of lake fill-drain cycles, and at least some elements of the associated basal hydrology system, may not be that important for modelling Thwaites Glacier's contribution to sea level on decadal to centennial timescales.

## 5 Conclusions

We document the temporal change in velocity and elevation far from the grounding zone in response to the steepening of

Thwaites Glacier and three distinct systems of active lakes: one on the main Thwaites Glacier trunk, another in the western shear margin of Haynes Glacier, and a third in the westernmost tributary of Thwaites. At the LTHW GNSS site, over one hundred kilometres from the grounding line, ice velocity has accelerated at a nearly constant rate over the last decade. This background acceleration was interrupted in 2012 by the connected drainage of lakes $Thw_{124}$, $Thw_{142}$ and $Thw_{170}$, and, in 2017, by the partial filling of $Thw_{124}$ via drainage of $Thw_{142}$ and $Thw_{170}$. Our observations suggest that the transport of ~2 cubic

kilometres of water beneath Thwaites Glacier, which represents approximately half the annual basal meltwater production for the entire Thwaites catchment (Joughin et al., 2009), has only a small and transient effect on glacier speed relative to ongoing thinning driven by ocean melt.

## Video Supplement

Supporting videos of surface elevation change and water routing are available at doi:10.5446/44023 and 10.5446/44035.



## Acknowledgments

The work was supported by NASA Cryospheric Sciences grants NNX16AM01G (KC) and NNX17AG54G (IJ and BS), the NASA sea-level change team grant 80NSSC17K0698 (KC), and the NSF-NERC International Thwaites Glacier Collaboration grant OPP-1738934 (KC and AH). GNSS data were provided by UNAVCO, NASA CDDIS, and SOPAC. We thank the POLENET team for maintaining the GPS sites. Logistical support was provided by Raytheon Polar Services, the New York Air National Guard, Kenn Borek Air, and the US Antarctic Support Contract. The authors declare no conflicts of interest.

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

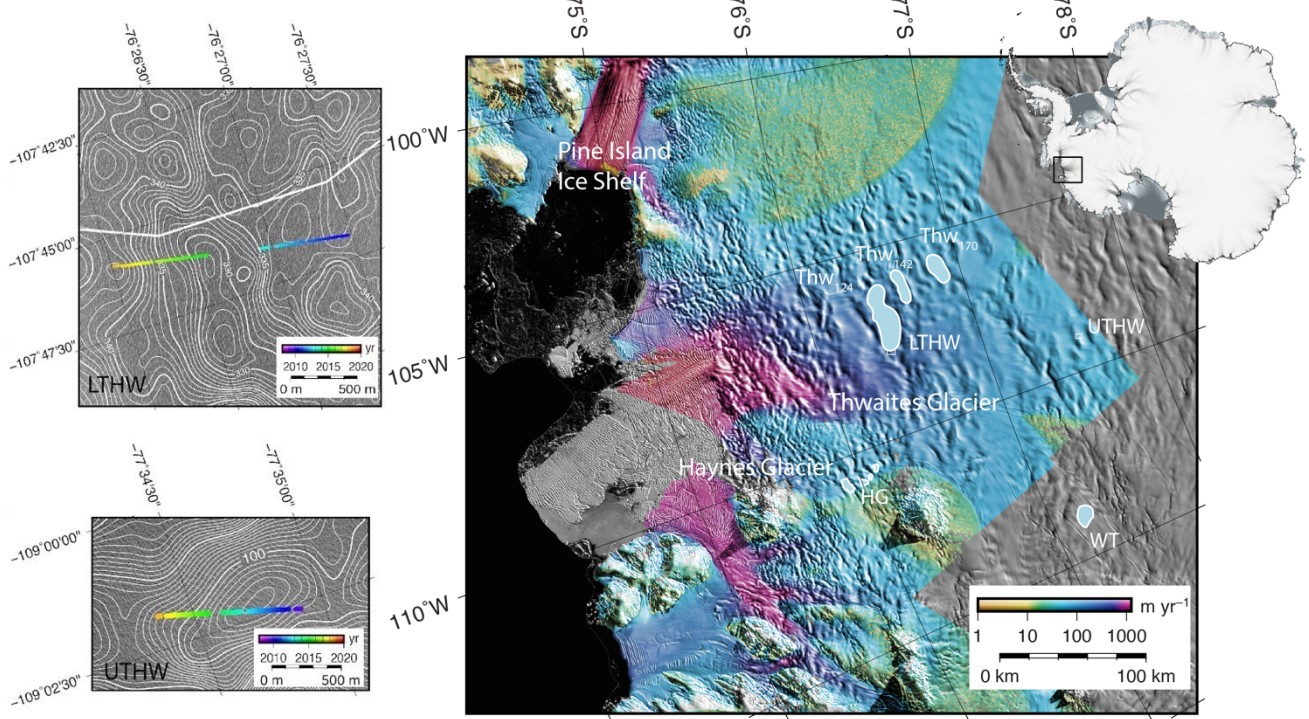


**Figure 1. Location map of Thwaites Glacier and subglacial Thwaites lakes. a) Ice speed (colour) plotted over Moderate Resolution Imaging Spectroradiometer (MODIS) image mosaic. Thwaites Glacier, Thwaites Lake 124 (THW$_{124}$) Thwaites Lake 142 (Thw$_{142}$), Thwaites Lake 170 (Thw$_{170}$), Haynes Glacier (HG) lake, Western Thwaites (WT) lake, and GNSS sites (LTHW and UTHW) are**
**labelled. b) LTHW and UTHW GNSS position plotted over time (colour) with contoured mean velocity.**



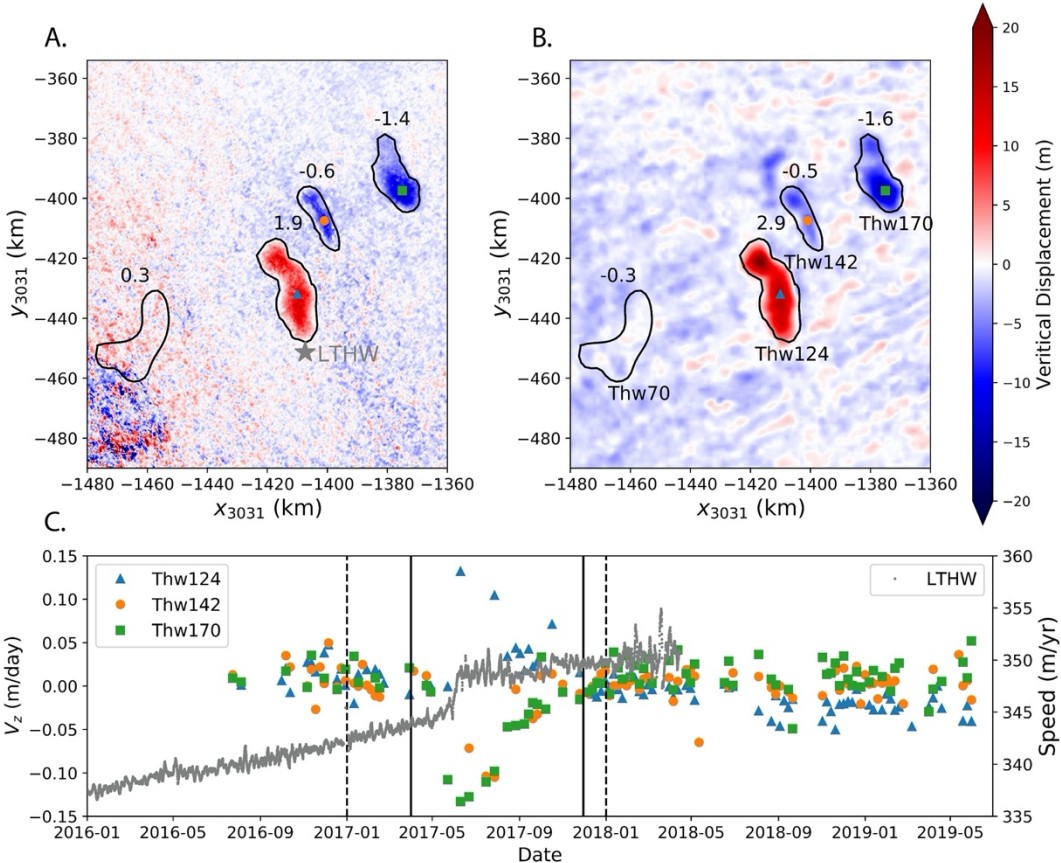

**Figure 2.** Surface elevation-change time series over the Thwaites Glacier lakes showing the 2017 drainage cascade from (A) vertical displacement computed from integrated vertical displacement rates (Vz) from Sentinel-1 SAR data and (B) swath-processed radar altimetry. (C) Time series of uplift rates ($V_z$) from SAR LOS results (left abscissa) and horizontal speed from GNSS observations (right abscissa). Solid lines represent period over which SAR vertical displacements ($V_z$) were integrated to produce the vertical displacements shown in panel A. Dotted lines represent the quarters of gridded CryoSat-2 data differenced to create panel B.


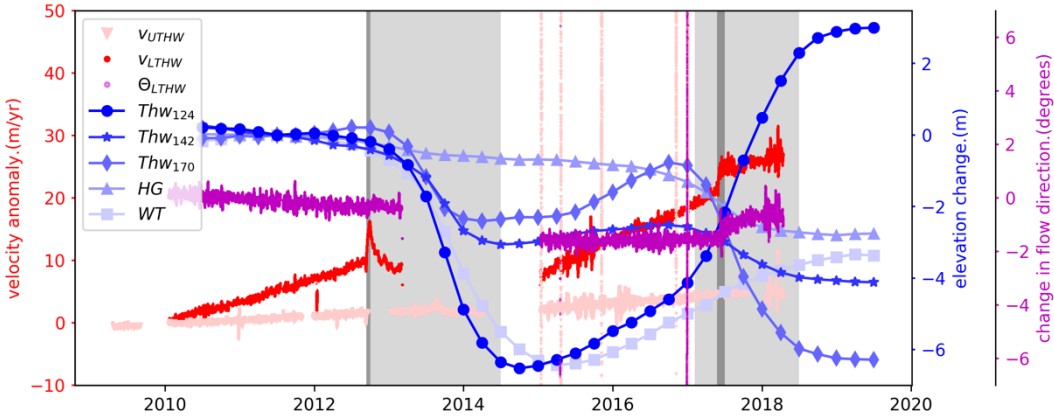


**Figure 3. Time series of CryoSat-2 lake elevation change averaged over each lake area and GNSS velocity anomalies at UTHW and LTHW corrected for advection using the Eulerian velocity products. Also plotted are LTHW GNSS direction change. The dark grey shaded periods indicate intervals when the LTHW GNSS accelerated significantly while the light grey periods indicate when**
**the lakes drain. When the largest lake fills in 2017, the LTHW GNSS closest to the lake accelerates and flows towards the lake.**
