# Peer review of "Brief Communication: Heterogenous thinning and subglacial lake activity on Thwaites Glacier, West Antarctica"

_The Cryosphere, 2020_

## Referee Comment (RC1) · Anonymous Referee #1 · 18 May 2020

General comments: This paper describes the behaviour of subglacial lakes in the Thwaites Glacier region. It describes an extension to an existing dataset with valuable new observations. It is well-suited for a 'brief communication' as it is timely and of relevance to ongoing research in this area. The authors use the new data to support the conclusion that subglacial lakes have a small effect on overall ice flow and conclude that their fill-drain cycles can be largely ignored when interpreting long-term trends due to the negligible effect on basal friction. The results are concisely presented.

Specific comments: Both the filling and draining of lakes have a small effect on the instantaneous velocity, but it seems that the long-term effect on the rate of acceleration

is more ambiguous. There is definitely a change in acceleration between 2010-12 and 2014-15. Is the authors' opinion that this is as a result of the lake drainage or driven by change elsewhere on the glacier? The lack of acceleration across the GNSS data gap in 2013-2014 followed by a faster rate of acceleration afterwards requires additional explanation. Contrary to the main conclusion, this overall long-lasting drop of ∼5% in velocity relative to the 2010-2012 trend may still have some importance in decadal trends. Despite these minor details the overall conclusions of the paper appear reasonable.

Technical comments: Fig 1: needs a/b labels

Fig 2: I would swap the y-axes for ease of reading, given the temporal distribution of the data.

Line 85: does this refer to the filling rate or the draining rate?

Line 95: what is the evidence that it is driven from upstream and not downstream?

Line 100: reword the sentence "This roughly..." for clarity

Line 135: reword the sentence "Enhanced lubrication..." for clarity

Supplement: I assume the figure at the bottom of page 5 is the panelled image referred to in S4? Perhaps consider renaming it to S5.

---

## Referee Comment (RC2) · Anonymous Referee #2 · 9 Jul 2020

While the presence of active subglacial lake systems in Greenland and Antarctica has been known for decades, the impact of the filling and draining of the lakes on the ice flow is still not well understood. This paper provides a comprehensive investigation using remote sensing observations and continuous GNSS monitoring on the Thwaites and Haynes glaciers in Antarctica, in a region that is undergoing rapid changes in ice dynamics. The paper is well written and presents excellent observational data sets combined with modeling subglacial water routing and basal friction estimation. The study demonstrates an innovative use of remote sensing, including the generation of high temporal resolution records of vertical displacement from Sentinel observations and ice sheet elevation from radar altimetry. The combined interpretation of the obser-

vations and the modeling results suggests that ice acceleration is not or only weakly sensitive to subglacial drainage, and, thus, the authors conclude that while the 2012 speed-up of the Thwaites Glacier trunk occurred shortly after the 2013 drainage event, it was due to enhanced sub-ice-shelf melt.

The study is worthy of publication and includes important results, but it still leaves some questions open. The authors lay out a convincing argument about the evolution of the subglacial conditions using reasonable assumptions, supported by previous work. However, the two GNSS stations provide only limited information for a basin-scale interpretation. For example, it is not clear how sensitive the locations of UTHW and LTHW are for changes in subglacial hydrology or diffusion thinning originating from the grounding line. Showing UTHW on S Fig.3 would help in the interpretation. Also, due to its position on the boundary of Lake Thw124, LTHW might be sensitive to complicated local processes that could even reduce the response to the drainage events.

Also, there are two questions that the manuscript could have answered:

1. Smith et al., 2017 hypothesized that lake drainage events would occur in 20-80 years periods. Do the authors have an explanation of the observed much shorter timescale (∼6 years). Also, the range of elevation change is increasing in time (Fig. 3). Could the shorter and more substantial variation indicate a rearrangement of the drainage system and a potential increase of its sensitivity to changing forcing?

2. The authors conclude that the speed-up of Thwaites glacier following the 2013 drainage event was due to increasing sub-ice melt rather than the subglacial lake drainage events. Does it mean that the two types of events (acceleration and drainage) not connected? Or could the drainage events be caused by slight changes in velocity/subglacial routing as the glacier started to speed up and thin?

Detailed comments:

Lines 36-37: I suggest to show Backer Island and Howard Nunatak on Fig. 1. I assume

that the distances are relative to one of the GNSS receivers – which one?

Line 39: Include reference for Savitzky-Golay filtered averages

Line 39-40: What is the time period for the Eulerian speed? Is it a mean velocity for a longer period or derived from a single SAR image pair?

Line 45: I assume that the component of motion in LOS direction was estimated by InSAR processing. Please include a reference.

Line 54: Add the word "solid" before vertical bars to distinguish from the dashed vertical bars.

Lines 65-66: Include explanation for E (expected value)

Line 65-67: This sentence is confusing. What is the "respectively" refer to?

Lines 83-84: The western Thwaites tributary and Haynes Glacier Lakes appears to be switched, according to the text, the Thwaites tributary (WT) has a large drainage event, while Fig. S2 shows the larger drainage for the Haynes Glacier lakes.

Lines 99-100: It is not clear what different average fill rates refer to. For example, ∼0.16 km3/yr appear to refer to the subglacial routing (Fig. S3), but the next sentence mentions the same estimate with a different value.

Line 135: LTHW is not shown in Fig. S3.

Figures:

The names of the lakes should be shown in the same way everywhere. Currently, both THW124 and Thw124, etc. are used.

Figure 1 caption: include the date (period) of the MODIS mosaic and the SAR velocity

Figure 2 caption: include projection – I assume it is EPSG 3031. Including a verbal description of the different symbols would make it easier to understand the figure, e.g., "from SAR LOS (colored dots, left abscissa or axis, locations marked in panels A and

B).

Figure 3 caption: again, description of the symbols in the caption would be helpful, especially for the symbol showing the angle, e.g., "Also plotted the LTHW GNSS station direction change (purple dots)." Should include a reference to Fig. 1 for finding the locations and abbreviations. Finally, which direction is the direction given? Clockwise or counterclockwise?

[Figure]

---

## Author Comment (AC2) · 10 Oct 2020

*Interactive comment on* "Brief Communication: Heterogenous thinning and subglacial lake activity observed on Thwaites Glacier, West Antarctica"

**Referee: Anonymous Referee #2**

We thank the reviewer for detailed comments with constructive criticism.

Our responses are marked as follows:

*Reviewer Comment (blue italic)*
Response (black)
New or changed text (red)

*While the presence of active subglacial lake systems in Greenland and Antarctica has been known for decades, the impact of the filling and draining of the lakes on the ice flow is still not well understood. This paper provides a comprehensive investigation using remote sensing observations and continuous GNSS monitoring on the Thwaites and Haynes glaciers in Antarctica, in a region that is undergoing rapid changes in ice dynamics. The paper is well written and presents excellent observational data sets combined with modeling subglacial water routing and basal friction estimation. The study demonstrates an innovative use of remote sensing, including the generation of high temporal resolution records of vertical displacement from Sentinel observations and ice sheet elevation from radar altimetry. The combined interpretation of the observations and the modeling results suggests that ice acceleration is not or only weakly sensitive to subglacial drainage, and, thus, the authors conclude that while the 2012 speed-up of the Thwaites Glacier trunk occurred shortly after the 2013 drainage event, it was due to enhanced sub-ice-shelf melt. The study is worthy of publication and includes important results, but it still leaves some questions open. The authors lay out a convincing argument about the evolution of the subglacial conditions using reasonable assumptions, supported by previous work. However, the two GNSS stations provide only limited information for a basinscale interpretation. For example, it is not clear how sensitive the locations of UTHW and LTHW are for changes in subglacial hydrology or diffusion thinning originating from the grounding line. Showing UTHW on S Fig.3 would help in the interpretation.*

We thank Reviewer 2 for raising the concern that localized GNSS coverage of Thwaites Glacier limits the extension of these observations to basin-scale understanding. We note that these two GNSS sites are the only two long-term sites that have been deployed on Thwaites Glacier and while their spatially coverage is inherently limited, they offer temporal resolution that is valuable for examining rapid processes, including lake drainages. Although serendipitous, the placement of the LTHW site is especially valuable for observing $Thw_{124}$. The UTHW site was selected to investigate large changes in modeled basal shear stress, but it is place along a central flowline, and thus is reasonably well-positioned to sample inland acceleration and thinning. We have added the position of the GNSS sites to Figure S3. In conjunction with this change and the additional text below, we also use monthly Eulerian observations of the glacier's speed to show that spatially distributed changes in glacier slip due to subglacial lake activity are small relative

to the average velocity of the lakes before and after the 2017 drainage event (Fig S5). These observations complement the inversions before and after the lakes drain in 2017 and show no substantial difference in glacier speed due to lake drainage beyond the immediate vicinity of the lake (Fig. S4). These figures have been included in the supplement.

Line 132-135: Speed remained elevated at the LTHW site after $Thw_{124}$ stopped filling coinciding with a 2-degree shift in ice-flow direction to the grid-north (clockwise), toward $Thw_{124}$ (Fig. 3); however, this speed change is imperceptible in distributed velocity maps before and after $Thw_{124}$ filled in 2017 (Fig. S5).

Supplement Figure 3: Average water flux assuming static hydropotential and basal melt rates from Joughin et al. (2009). Supplement movie shows weak sensitivity for water rerouting as the glacier thins and the lakes fill and drain. The cumulative water fluxes ($km^3$/yr) into lakes $Thw_{124,142,170}$ are printed with each lake. Black star and square indicate sites of LTHW and UTHW GNSS.

*Also, due to its position on the boundary of Lake $Thw_{124}$, LTHW might be sensitive to complicated local processes that could even reduce the response to the drainage events.*

We recognize that hydraulic flexure and viscoelastic response of ice near LTHW during and after lake drainage may contribute to complicated speed change we observe near the $Thw_{124}$ lake margin (see lines 168-172). We note here that the response time of significant elastic deformation is faster than the 10 day speed up and slow down we observe at the LTHW GNSS in 2012. For comparison, the ephemeral subglacial lakes in Greenland that form following supraglacial lake drainage cause changes in ice velocity that equilibrate over the course of a single day (Stevens et al. 2015).

*Also, there are two questions that the manuscript could have answered:*

*1. Smith et al., 2017 hypothesized that lake drainage events would occur in 20-80 years periods. Do the authors have an explanation of the observed much shorter timescale (~6 years).*
The shorter timescale of lake filling and draining is certainly interesting. Because the timeseries is short and we do not know the absolute volume of the lakes, the water volume upstream of the lakes, or the connectivity of upstream water bodies to the Thwaites lakes, we do not feel we can say with confidence whether the lake drainages are fully inconsistent with the 20-80 year average period proposed by Smith et al. (2017). We also note that changes in lake storage capacity and lake drainage reoccurrence interval are also not always directly related, noted here for Thwaites but also documented on the Siple Coast (Siegfried et al., 2014; 2016), and indicate likelihood for non-constant recharge periods (see lines 168-172 of the revised manuscript).

*Also, the range of elevation change is increasing in time (Fig. 3).*
This statement is true for the largest Thwaites lake ($Thw_{124}$), which notably lies downstream of the other active Thwaites lakes. This observation is also consistent with subglacial lake activity observed along the Siple Coast (Siegfried et al., 2014; 2016). The apparent change in the storage capacity of $Thw_{124}$ suggests that the lake volume and timing of lake drainage depends on the

dynamics of subglacial water flow between the lakes (initial effective pressure, conduit morphology, ice velocity, etc.) and hydraulic disconnection mechanisms in addition to the static hydropotential (lines 168-172 of the revised manuscript).

*Could the shorter and more substantial variation indicate a rearrangement of the drainage system and a potential increase of its sensitivity to changing forcing?*
In the movie supplement, we show a time series of the thinning observations (Movie SV1; doi:10.5446/44023) and changes in water routing affected by on-going thinning (Moive SV2; doi:10.5446/44035). The time-evolving Shreve (1972) hydropotential indicates that water routing is relatively insensitive to the progressive thinning that occurred over the time series of observations presented in this study. We do not observed sensitivity of hydraulic flowpaths to minor elevation changes (<15 m) that has been suggested to occur elsewhere in Antarctica (Wright et al., 2008). The Shreve hydropotential changes linearly with the overburden pressure, which decreases as Thwaites glacier continues to thin. Thus, although the effects of water routing are minimal, the ongoing thinning will likely change the storage capacity of the lakes and the characteristic fill-drain frequency, and we are doing work now to further understand these changes.

*2. The authors conclude that the speed-up of Thwaites glacier following the 2013 drainage event was due to increasing sub-ice melt rather than the subglacial lake drainage events. Does it mean that the two types of events (acceleration and drainage) not connected? Or could the drainage events be caused by slight changes in velocity/subglacial routing as the glacier started to speed up and thin?*
The acceleration and drainage cannot conclusively be linked; however, the acceleration due to the mechanics of lake drainage from the 2012-2013 GNSS record appear to be larger than the background rate of acceleration we attribute to thinning near the grounding line. This suggests that the dynamics of the lake filling and draining may locally and temporally supersede the effects of acceleration due to thinning upon initiation of lake filling or draining. We state this in lines 175-177 and also recognize that these fluctuations are of insufficient magnitude and duration to affect long-term trends.

*Lines 36-37: I suggest to show Backer Island and Howard Nunatak on Fig. 1. I assume that the distances are relative to one of the GNSS receivers – which one?*
We have modified Figure 1 to include Backer Island and Howard Nunatak. We have included the distance from Howard Nunatak to both sites as the Howard Nunatak reference station was used to kinematically process the on-ice GNSS positions posted in all of the figures and described in the text. See further changes below.

*Line 39: Include reference for Savitzky-Golay filtered averages*
The reference for the Savitzky-Golay filter was considered for this text; however, we are only allowed 20 citations. We request that we are allowed more citations and include the Svitzy-Golay citation as follows.

Line 39-40: We then constructed velocity time series from these geodetic solutions using 3-day Savitzky-Golay filtered moving averages (Press et al. 2007).

Press, W. H., S. A. Teukolsky, W. T. Vetterling, and B. P. Flannery (2007), Numerical Recipes, 3rd ed., Cambridge Univ. Press, Cambridge, U. K.

*Line 39-40: What is the time period for the Eulerian speed? Is it a mean velocity for a longer period or derived from a single SAR image pair?*
The time period of the Eulerian speed is a mean velocity product from 2015-2019, but excludes velocity maps sampled when significant vertical velocity change over the lakes affects the assumptions for distributed horizontal velocity. This explanation is now included in lines 40-41 and the Figure 1 text.

*Line 45: I assume that the component of motion in LOS direction was estimated by InSAR processing. Please include a reference*
The component of motion in LOS direction was estimated from SAR processing. We have added citation in lines 46-47 .

Line 46-47: We also computed the component of motion in the satellite line-of-sight (LOS) direction (Gray et al., 2005; Friedl et al., 2020).

*Line 54: Add the word "solid" before vertical bars to distinguish from the dashed vertical bars*
Sentence has been modified as suggested.

Line 53-56: To more tightly constrain the timing of the drainage events, we spatially interpolated the time series of Sentinel-1 derived Vz to fill gaps in coverage and integrated the result during a period of filling/draining (see solid vertical bars in Fig. 2c) to produce estimates of net uplift and subsidence shown in Figure 2a.

*Lines 65-66: Include explanation for E (expected value)*
The expected elevation statistics are defined in Smith et al. (2017); however, we agree that these statistics should be described again to provide context for readers without consulting another paper.

Line 67-70: The elevation statistics, $E\left(\frac{\partial^2 z_0}{\partial x^2}\right)$, $E\left(\frac{\partial^3 z_0}{\partial x^2 \partial t}\right)$, and $E\left(\frac{\partial^2 z_0}{\partial t^2}\right)$, represent expected values for spatial and temporal derivatives of the reference elevation model, $z_0$, and the time dependent height-change field, $z$.

*Line 65-67: This sentence is confusing. What is the "respectively" refer to?*
This sentence aimed to state the elevation statistics and compare these values with those previously used to compute elevation change on Thwaites Glacier in Smith et al. (2017). The word respectively is used to link the factor change in expected value to the associated elevation statistic ($E\left(\frac{\partial^2 z_0}{\partial x^2}\right)$ was changed by a factor of 5 and $E\left(\frac{\partial^3 z_0}{\partial x^2 \partial t}\right)$ was changed by a factor of 10 relative to Smith et al. (2017). We have attempted to reword for clarity.

Line 68-71: The values chosen for this study are $E\left(\frac{\partial^2 z_0}{\partial x^2}\right) = 6.7 \times 10^{-8}\ m^{-2}$, $E\left(\frac{\partial^3 z_0}{\partial x^2 \partial t}\right) = 6 \times 10^{-9}\ m^{-2} yr^{-1}$, and $E\left(\frac{\partial^2 z_0}{\partial t^2}\right) = 1.0 myr^{-2}$, and tighten the spatial variations in the least square's elevation-change time series, $E\left(\frac{\partial^2 z_0}{\partial x^2}\right)$, $E\left(\frac{\partial^3 z_0}{\partial x^2 \partial t}\right)$ compared to the original Smith et al. (2017) paper by factors of 5 and 10, respectively.

*Lines 83-84: The western Thwaites tributary and Haynes Glacier Lakes appears to be switched, according to the text, the Thwaites tributary (WT) has a large drainage event, while Fig. S2 shows the larger drainage for the Haynes Glacier lakes.*
We appreciate this correction. See changes to Figure S2.

*Lines 99-100: It is not clear what different average fill rates refer to. For example, ~0.16 km3/yr appear to refer to the subglacial routing (Fig. S3), but the next sentence mentions the same estimate with a different value.*
We have changed the second and third sentence in this paragraph to more accurately convey the origin of the volume change rates.

Line 108-111: From the altimetric observations of the $Thw_{170}$ fill cycle, the average fill rate is ~0.16km³/yr (Fig. S3). This agrees with the fill rate (~0.14km³/yr) we calculate by routing inferred basal meltwater production (Joughin et al., 2009) down the glaciostatic hydropotential gradient (Shreve, 1972) into $Thw_{170}$, but requires inflow of all melt water produced upstream into the $Thw_{170}$ lake basin (Fig. S3).

*Line 135: LTHW is not shown in Fig. S3.*
We thank referee 2 for noticing this omission. LTHW is now included.

*Figures:*
*The names of the lakes should be shown in the same way everywhere. Currently, both THW124 and Thw124, etc. are used. Figure 1 caption: include the date (period) of the MODIS mosaic and the SAR velocity*
We thank referee 2 for catching the inconsistent labeling. The lakes are now marked consistently throughout the text and figures. We have also added the dates for the SAR averaged velocity data and a citation for the MODIS mosaic (Haran et al., 2014). The Figure 1 caption now reads:

Figure 1. Location map of Thwaites Glacier and subglacial Thwaites lakes. (A) Average ice speed between 2015-2019 omitting period when lakes were active (colour) plotted over Moderate Resolution Imaging Spectroradiometer (MODIS) image mosaic (Haran et al., 2014). Thwaites Glacier, Thwaites Lake 124 ($Thw_{124}$) Thwaites Lake 142 ($Thw_{142}$), Thwaites Lake 170 ($Thw_{170}$), Haynes Glacier (HG) lake, Western Thwaites (WT) lake, and GNSS sites (LTHW and UTHW) are labelled. Thwaites lakes are named by their approximate distance from the grounding line. (B) LTHW and (C) UTHW GNSS position plotted over time (colour) with contoured mean velocity between 2015-2019.

*Figure 2 caption: include projection – I assume it is EPSG 3031. Including a verbal description of the different symbols would make it easier to understand the figure, e.g., "from SAR LOS (colored dots, left abscissa or axis, locations marked in panels A and C3 B).*

The projection is EPSG:3031 (polar stereographic centered at the South Pole, with latitude of true scale at 71ºS and the central meridian is the prime meridian). We have changed the figure 2 caption to the text below.

Figure 2. Surface elevation-change time series over the Thwaites Glacier lakes showing the 2017 drainage cascade from (A) vertical displacement computed from integrated vertical displacement rates (Vz) from Sentinel-1 SAR data and (B) swath-processed radar altimetry in a polar stereographic projection (EPSG:3031). Water volume ($km^3$) associated with observed vertical displacement is labelled for each lake. (C) Time series of uplift rates (Vz) from SAR LOS results (coloured dots, left abscissa; locations marked in panels A and B and horizontal speed from GNSS observations (right abscissa). Solid lines represent period over which SAR vertical displacements (Vz) were integrated to produce the vertical displacements shown in panel A. Dotted lines represent the quarters of gridded CryoSat-2 data differenced to create panel B.

*Figure 3 caption: again, description of the symbols in the caption would be helpful, especially for the symbol showing the angle, e.g., "Also plotted the LTHW GNSS station direction change (purple dots)." Should include a reference to Fig. 1 for finding the locations and abbreviations. Finally, which direction is the direction given? Clockwise or counterclockwise?*

The direction is clockwise, so shifts to the north relative to the westward flow direction. See additions to figure caption below.

Figure 3. Time series of GNSS velocity anomalies at UTHW and LTHW corrected for advection using the Eulerian velocity products and CryoSat-2 lake elevation change averaged over each lake area . See *Fig. 1* for site locations and abbreviations. Also plotted are LTHW GNSS clockwise direction change relative to 2010 flow direction (purple). The dark grey shaded periods indicate intervals when the LTHW GNSS accelerated significantly (99% confidence) while the light grey periods indicate when the lakes drain. When the largest lake fills in 2017, the LTHW GNSS closest to the lake accelerates and flows towards the lake.

**References**

Stevens, L., Behn, M., McGuire, J. *et al.* Greenland supraglacial lake drainages triggered by hydrologically induced basal slip. *Nature* 522, 73–76 (2015). https://doi.org/10.1038/nature14480

---

## Author Response (AR1)

Referee 1 suggests that the filling and draining of lakes has a more ambiguous effect on the rate of long-term acceleration across Thwaites Glacier than we conclude in the paper. Negative acceleration in ice motion near Thw124 across the GNSS data gap between 2013-2014 followed by the faster rate of acceleration when GNSS begin telemetering in 2015 is the combined signal of lake activity and ongoing thinning. Acceleration that would accompany progressive thinning over this period is convolved with changes in the stress state due to lake dynamics and the subglacial hydrology system and cannot easily be disentangled without sub-annual distributed velocity information, which are not available for this time period (2010-2014). We present distributed velocity data during the second lake drainage event in this manuscript. Before and after the upper Thwaites lakes drained, filling the largest Thwaites lake, the velocity of the ice in the vicinity of the lakes does not discernably change (Fig. S3). We note that this drainage cycle is ongoing and that Thw124 has not yet drained since filling in 2017. Although repeated active lake drainage events are somewhat similar elsewhere where observations do exist (Siegfried et al., 2014, 2016), they are not identical. As we have no distributed velocity data that spans the full upper Thwaites lake drainage sequence, we prefer not to speculate on velocity expression of the first series of lake drainages besides noting that it is due to both progressive thinning and

subglacial lake activity. These thoughts are more concisely presented in lines 169-175 of the revised manuscript.

**The lack of acceleration across the GNSS data gap in 2013-2014 followed by a faster rate of acceleration afterwards requires additional explanation.**

The lack of acceleration across the GNSS data gap could be the result of many different speed-up and slow down scenarios driven by combined lake activity, localized viscoelastic ice response to the drainage, and basin-wide acceleration. As we stated above, additional sub-annual distributed velocity data are not available for this time period to evaluate hypothetical scenarios. Due to the GNSS receiver power failure, we only know the average change in speed over this period and the acceleration after the GNSS began telemetering again. We do have distributed velocity data from the second lake drain-fill cycle in 2017, which indicates that the resistance field in the boundary of the lake does not discernably change before and after the lakes drain. This observation later in the timeseries coupled with the observed speedup near the grounding zone in 2013-2014 measured with feature tracked remote sensing imagery (Fig. S6 from Smith et al., 2017) suggests that the acceleration after 2013-2014 may be the catchment interior responding to thinning initiated at the grounding line in response to increasing basal-melt rates driven by ocean warming in the Amundsen Sea (Christianson et al., 2016); however, without distributed velocity data at sub-annual temporal resolution over the lake during this period, it would be conjectural to attribute the stagnation and acceleration to one series of mechanisms. We summarize these thoughts in lines 168-172 of the revised manuscript.

These new observations suggest that the observed speed-up at the grounding zone of the main trunk of Thwaites Glacier following the 2013 drainage (Smith et al., 2017) was associated with warming ocean conditions following anomalous Amundsen Sea wide ocean cooling from 2012-2013 (Christianson et al., 2016). These warm ocean conditions likely enhanced sub-ice-shelf melt and led to increased ungrounding and acceleration.

**Contrary to the main conclusion, this overall long-lasting drop of $\sim$ 5% in velocity relative to the 2010-2012 trend may still have some importance in decadal trends. Despite these minor details the overall conclusions of the paper appear reasonable.**

When considering the impact of lake activity on ice motion there are two process timescales: (1) a fast response that includes the viscoelastic response of the ice-sheet and (2) a slow viscous response toward a new equilibrium. The fast response can only be measured with GNSS, while the slow response changes the equilibrium geometry and speed, which we measure with satellite remote sensing. Lake activity can affect both of these modes of ice sheet response by flexing the ice sheet, during rapid lake filling and draining, and dewatering the bed, which can change the local basal resistance on the timescale of lake filling and draining (years to decades, Smith et al., 2017). Furthermore, the effects of lake activity on elevation and velocity change appear to be quite local (see Fig. S3). During the 2013-2014 data gap, the acceleration at the LTHW station (at the boundary of the draining Thw124 lake) changed from  $\sim 3m/yr^2$  (average acceleration before the drainage event) to  $\sim 0m/yr^2$ , but this fluctuation represents less than 3% of the total velocity signal and is much smaller than the velocity variability driven by changes near the grounding line (Miles et al., 2020). Due to the lack of distributed highly temporally resolved velocity data during the 2012-2013 drainage period, we cannot determine the spatial extent of these changes,

limiting our ability to attribute speed changes to local (lake drainage) or broader (basin-wide thinning) processes. The fact that the GNSS located at the boundary of  $Thw_{124}$  was only minorly affected by change in slip (~1% speed increase) when the lake filled in 2017 suggests that ice flexure and changes in the sliding speed due to lake fill and drain cycles have a spatially limited affect on ice motion. The snap-shot inversions over the lakes before and after the system drained in 2017 are consistent with this hypothesis (Fig. S5). The shear stress inside the lake boundaries does not change significantly, indicating that shear stress is low inside the lake area regardless of lake level (inversions were done before and after the 2017 Thwaites lakes drainage event). Therefore, we conclude that the lakes have only minor and localized effects on ice dynamics. These effects are far too limited in area and magnitude to affect basin wide velocity trends. We present these thoughts more compactly in lines 168-172 of the revised manuscript.

**Technical comments:**

**Fig 1: needs a/b labels**

Figure 1 has been modified to include labels A, B, and the addition of a C subpanel to distinguish the LTHW and UTHW GNSS sites. We thank the reviewer for catching this omission. See modified figure text below. Also, note change to Thwaites Lake identifier Thw124, which is in lower case to be consistent with literature (Smith et al., 2017) and references throughout the text. We have also added the dates for the SAR average velocity field and a citation for the MODIS mosaic.

Figure 1. Location map of Thwaites Glacier and subglacial Thwaites lakes. (A) Average ice speed between 2015-2019 omitting period when lakes were active (colour) plotted over Moderate Resolution Imaging Spectroradiometer (MODIS) image mosaic (Haran et al., 2014). Thwaites Glacier, Thwaites Lake 124 (Thw124) Thwaites Lake 142 (Thw142), Thwaites Lake 170 (Thw170), Haynes Glacier (HG) lake, Western Thwaites (WT) lake, and GNSS sites (LTHW and UTHW) are labelled. Thwaites lakes are named by their approximate distance from the grounding line. (B) LTHW and (C) UTHW GNSS position plotted over time (colour) with contoured mean velocity between 2015-2019.

**Fig 2: I would swap the y-axes for ease of reading, given the temporal distribution of the data.**

We have kept the axes as they were first plotted to emphasize the vertical velocity change we measure in SAR LOS data, which was plotted in the primary axis position with the observed speed change plotted in the twin axis position. This also matches the axes plotted in the supplement for the Haynes Glacier lakes. We think this allows easier synthesis with the spatial extent of the vertical velocity change plotted in panels A and B of this figure for the time periods shown in panel C. We have changed the figure 2 text to better link the subplots and aid reader interpretation (see changes below).

Figure 2. Surface elevation-change time series over the Thwaites Glacier lakes showing the 2017 drainage cascade from (A) vertical displacement computed from integrated vertical displacement rates (Vz) from Sentinel-1 SAR data and (B) swath-processed radar altimetry in a polar stereographic projection (EPSG:3031). Water volume (km3) associated with observed vertical displacement is labelled for each lake. (C) Time series of uplift rates (Vz) from SAR LOS results (coloured dots, left abscissa; locations marked in panels A and B and horizontal speed from

GNSS observations (right abscissa). Solid lines represent period over which SAR vertical displacements (Vz) were integrated to produce the vertical displacements shown in panel A. Dotted lines represent the quarters of gridded CryoSat-2 data differenced to create panel B.

**Line* 85: *does this refer to the filling rate or the draining rate?**

Line 85 refers to the filling rate. See sentence (line 89 in revised manuscript) restructured for clarity below.

Line 89: The western Thwaites tributary lake (WT), however, fills significantly at a rate of 0.1km3/yr after draining in 2014.

*Line 95: what is the evidence that it is driven from upstream and not downstream?* The differences in static hydropotential between the lakes is too large for the connection to be driven by the downstream lakes. See Supplement Figure S4.

*Line 100: reword the sentence "This roughly..." for clarity* The sentence starting in line 100 has been changed to read:

Lines 105-107: This agrees with the fill rate (~0.14km3/yr) we calculate by routing basal meltwater production (Joughin et al., 2009) down the Shreve glaciostatic hydropotential gradient (Shreve, 1972) into Thw170, but requires inflow of all melt water produced upstream into the Thw170 lake basin (Fig. S3).

**Line 135: reword the sentence "Enhanced lubrication..." for clarity* Line 135 has been changed to read:**

Line 142-145: Enhanced lubrication outside the low-drag  $Thw_{124}$  basin as the lake begins to drain likely increases local slip and drives the subtle change in ice-flow direction that we observe in the austral winter of 2012 before the peak drainage in 2013, when flow direction shifts back to the mean flow direction between 2010-2012.

Supplement: I assume the figure at the bottom of page 5 is the panelled image referred to in S4 (now S5)? Perhaps consider renaming it to S5.

The Supplement figure S5 has been modified into two panels. The primary panel has been labeled A, and the figure showing the difference in the inferred friction proxy has been labeled S5 B. See change in description below.

Supplement Figure 4: Static inversion for basal resistance field for 2017 catchment geometry (A) before the Haynes Glacier and Thwaites Glacier drainage events and (B) difference in inferred basal resistance between two static inversions from 2017 and 2018 (before and after the 2017 drainage cascade) for Thw124,142,170.

**References**

Miles, B., Stokes, C., Jenkins, A., Jordan, J., Jamieson, S., & Gudmundsson, G. (2020). Intermittent structural weakening and acceleration of the Thwaites Glacier Tongue between 2000 and 2018. Journal of Glaciology, 66(257), 485-495. doi:10.1017/jog.2020.20 *Interactive comment on* "Brief Communication: Heterogenous thinning and subglacial lake activity observed on Thwaites Glacier, West Antarctica"

**Referee: Anonymous Referee #2** Received and published: 9 July 2020

We thank the reviewer for detailed comments with constructive criticism.

Our responses are marked as follows:

Reviewer Comment (blue italic) Response (black) New or changed text (red)

While the presence of active subglacial lake systems in Greenland and Antarctica has been known for decades, the impact of the filling and draining of the lakes on the ice flow is still not well understood. This paper provides a comprehensive investigation using remote sensing observations and continuous GNSS monitoring on the Thwaites and Haynes glaciers in Antarctica, in a region that is undergoing rapid changes in ice dynamics. The paper is well written and presents excellent observational data sets combined with modeling subglacial water routing and basal friction estimation. The study demonstrates an innovative use of remote sensing, including the generation of high temporal resolution records of vertical displacement from Sentinel observations and ice sheet elevation from radar altimetry. The combined interpretation of the observations and the modeling results suggests that ice acceleration is not or only weakly sensitive to subglacial drainage, and, thus, the authors conclude that while the 2012 speed-up of the Thwaites Glacier trunk occurred shortly after the 2013 drainage event, it was due to enhanced sub-ice-shelf melt. The study is worthy of publication and includes important results, but it still leaves some questions open. The authors lay out a convincing argument about the evolution of the subglacial conditions using reasonable assumptions, supported by previous work. However, the two GNSS stations provide only limited information for a basinscale interpretation. For example, it is not clear how sensitive the locations of UTHW and LTHW are for changes in subglacial hydrology or diffusion thinning originating from the grounding line. Showing UTHW on S Fig.3 would help in the interpretation.

We thank Reviewer 2 for raising the concern that localized GNSS coverage of Thwaites Glacier limits the extension of these observations to basin-scale understanding. We note that these two GNSS sites are the only two long-term sites that have been deployed on Thwaites Glacier and while their spatially coverage is inherently limited, they offer temporal resolution that is valuable for examining rapid processes, including lake drainages. Although serendipitous, the placement of the LTHW site is especially valuable for observing Thw124. The UTHW site was selected to investigate large changes in modeled basal shear stress, but it is place along a central flowline, and thus is reasonably well-positioned to sample inland acceleration and thinning. We have added the position of the GNSS sites to Figure S3. In conjunction with this change and the additional text below, we also use monthly Eulerian observations of the glacier's speed to show that spatially distributed changes in glacier slip due to subglacial lake activity are small relative

to the average velocity of the lakes before and after the 2017 drainage event (Fig S5). These observations complement the inversions before and after the lakes drain in 2017 and show no substantial difference in glacier speed due to lake drainage beyond the immediate vicinity of the lake (Fig. S4). These figures have been included in the supplement.

Line 132-135: Speed remained elevated at the LTHW site after  $Thw_{124}$  stopped filling coinciding with a 2-degree shift in ice-flow direction to the grid-north (clockwise), toward  $Thw_{124}$  (Fig. 3); however, this speed change is imperceptible in distributed velocity maps before and after  $Thw_{124}$  filled in 2017 (Fig. S5).

Supplement Figure 3: Average water flux assuming static hydropotential and basal melt rates from Joughin et al. (2009). Supplement movie shows weak sensitivity for water rerouting as the glacier thins and the lakes fill and drain. The cumulative water fluxes (km3/yr) into lakes Thw124,142,170 are printed with each lake. Black star and square indicate sites of LTHW and UTHW GNSS.

**Also, due to its position on the boundary of Lake $Thw_{124}$ , LTHW might be sensitive to complicated local processes that could even reduce the response to the drainage events.**

We recognize that hydraulic flexure and viscoelastic response of ice near LTHW during and after lake drainage may contribute to complicated speed change we observe near the Thw124 lake margin (see lines 168-172). We note here that the response time of significant elastic deformation is faster than the 10 day speed up and slow down we observe at the LTHW GNSS in 2012. For comparison, the ephemeral subglacial lakes in Greenland that form following supraglacial lake drainage cause changes in ice velocity that equilibrate over the course of a single day (Stevens et al. 2015).

**Also, there are two questions that the manuscript could have answered:**

**1. Smith et al., 2017 hypothesized that lake drainage events would occur in 20-80 years periods. Do the authors have an explanation of the observed much shorter timescale (~6 years).**

The shorter timescale of lake filling and draining is certainly interesting. Because the timeseries is short and we do not know the absolute volume of the lakes, the water volume upstream of the lakes, or the connectivity of upstream water bodies to the Thwaites lakes, we do not feel we can say with confidence whether the lake drainages are fully inconsistent with the 20-80 year average period proposed by Smith et al. (2017). We also note that changes in lake storage capacity and lake drainage reoccurrence interval are also not always directly related, noted here for Thwaites but also documented on the Siple Coast (Siegfried et al., 2014; 2016), and indicate likelihood for non-constant recharge periods (see lines 168-172 of the revised manuscript).

**Also, the range of elevation change is increasing in time (Fig. 3).**

This statement is true for the largest Thwaites lake (Thw124), which notably lies downstream of the other active Thwaites lakes. This observation is also consistent with subglacial lake activity observed along the Siple Coast (Siegfried et al., 2014; 2016). The apparent change in the storage capacity of Thw124suggests that the lake volume and timing of lake drainage depends on the

dynamics of subglacial water flow between the lakes (initial effective pressure, conduit morphology, ice velocity, etc.) and hydraulic disconnection mechanisms in addition to the static hydropotential (lines 168-172 of the revised manuscript).

**Could the shorter and more substantial variation indicate a rearrangement of the drainage system and a potential increase of its sensitivity to changing forcing?**

In the movie supplement, we show a time series of the thinning observations (Movie SV1; doi:10.5446/44023) and changes in water routing affected by on-going thinning (Moive SV2; doi:10.5446/44035). The time-evolving Shreve (1972) hydropotential indicates that water routing is relatively insensitive to the progressive thinning that occurred over the time series of observations presented in this study. We do not observed sensitivity of hydraulic flowpaths to minor elevation changes (

**Line 135: LTHW is not shown in Fig. S3.**

We thank referee 2 for noticing this omission. LTHW is now included.

**Figures:**

The names of the lakes should be shown in the same way everywhere. Currently, both THW124 and Thw124, etc. are used. Figure 1 caption: include the date (period) of the MODIS mosaic and the SAR velocity

We thank referee 2 for catching the inconsistent labeling. The lakes are now marked consistently throughout the text and figures. We have also added the dates for the SAR averaged velocity data and a citation for the MODIS mosaic (Haran et al., 2014). The Figure 1 caption now reads:

Figure 1. Location map of Thwaites Glacier and subglacial Thwaites lakes. (A) Average ice speed between 2015-2019 omitting period when lakes were active (colour) plotted over Moderate Resolution Imaging Spectroradiometer (MODIS) image mosaic (Haran et al., 2014). Thwaites Glacier, Thwaites Lake 124 (Thw124) Thwaites Lake 142 (Thw142), Thwaites Lake 170 (Thw170), Haynes Glacier (HG) lake, Western Thwaites (WT) lake, and GNSS sites (LTHW and UTHW) are labelled. Thwaites lakes are named by their approximate distance from the grounding line. (B) LTHW and (C) UTHW GNSS position plotted over time (colour) with contoured mean velocity between 2015-2019.

Figure 2 caption: include projection – I assume it is EPSG 3031. Including a verbal description of the different symbols would make it easier to understand the figure, e.g., "from SAR LOS (colored dots, left abscissa or axis, locations marked in panels A and C3 B).

The projection is EPSG:3031 (polar stereographic centered at the South Pole, with latitude of true scale at 71°S and the central meridian is the prime meridian). We have changed the figure 2 caption to the text below.

Figure 2. Surface elevation-change time series over the Thwaites Glacier lakes showing the 2017 drainage cascade from (A) vertical displacement computed from integrated vertical displacement rates (Vz) from Sentinel-1 SAR data and (B) swath-processed radar altimetry in a polar stereographic projection (EPSG:3031). Water volume (km3) associated with observed vertical displacement is labelled for each lake. (C) Time series of uplift rates (Vz) from SAR LOS results (coloured dots, left abscissa; locations marked in panels A and B and horizontal speed from GNSS observations (right abscissa). Solid lines represent period over which SAR vertical displacements (Vz) were integrated to produce the vertical displacements shown in panel A. Dotted lines represent the quarters of gridded CryoSat-2 data differenced to create panel B.

Figure 3 caption: again, description of the symbols in the caption would be helpful, especially for the symbol showing the angle, e.g., "Also plotted the LTHW GNSS station direction change (purple dots)." Should include a reference to Fig. 1 for finding the locations and abbreviations. Finally, which direction is the direction given? Clockwise or counterclockwise? The direction is clockwise, so shifts to the north relative to the westward flow direction. See additions to figure caption below.

Figure 3. Time series of GNSS velocity anomalies at UTHW and LTHW corrected for advection using the Eulerian velocity products and CryoSat-2 lake elevation change averaged over each lake area . See *Fig. 1* for site locations and abbreviations. Also plotted are LTHW GNSS clockwise direction change relative to 2010 flow direction (purple). The dark grey shaded periods indicate intervals when the LTHW GNSS accelerated significantly (99% confidence) while the light grey periods indicate when the lakes drain. When the largest lake fills in 2017, the LTHW GNSS closest to the lake accelerates and flows towards the lake.

1Department of Earth and Space Sciences, University of Washington, Seattle, 98115, United States of America 2Applied Physics Laboratory, University of Washington, 98115, United States of America

Correspondence to: Andrew O. Hoffman (hoffmaao@uw.edu)

[revised manuscript text omitted]

Commented [AOH2]: Added "Glacier"
Commented [AOH3]: Added to satisfy reviewer two cmments.
Commented [AOH4]: Added citation for Savistzky-Golay filter.

Commented [AOH5]: Added the Friedl citation.

**Commented [AOH6]:** This sentence was changed to include solid in response to reviewer 2.

**2.2 Elevation and Lake Volume Change**

We also extended the previous time series of ESA CryoSat-2 radar altimetry data (Smith et al., 2017) through austral winter 2019, as shown in Figures 2b and 3. Elevation models were derived by fitting surfaces of elevation change to CryoSat-2 swath-

- 60 processed elevation retrievals and points-of-closest-approach relative to a reference elevation model from the first quarter of 2011 (Smith et al., 2017).] The fitting procedure minimized an objective functional that considered data misfit, spatial gradients in the constructed reference elevation model, elevation-change rate fields, temporal gradients in elevation-change rate, and the magnitude of model bias parameters. In this scheme, three expected elevation statistics are used to choose weight parameters that regularize the least-squares fit. The elevation statistics,  $E\left(\frac{\partial^2 z_0}{\partial x^2}\right)$ ,  $E\left(\frac{\partial^3 z}{\partial x^2 \partial t}\right)$ , and  $E\left(\frac{\partial^2 z}{\partial t^2}\right)$ , represent expected values for 65 spatial and temporal derivatives of the reference elevation model,  $z_0$ , and the time dependent height-change field, z. The values chosen for this study are  $E\left(\frac{\partial^2 z_0}{\partial x^2}\right) = 6.7 \times 10^{-8}$ ,  $E\left(\frac{\partial^3 z}{\partial x^2 \partial t}\right) = 6 \times 10^{-9}$  myr-1, and  $E\left(\frac{\partial^2 z}{\partial t^2}\right) = 1.0$ m2yr-2, and tighten the spatial variations in the least-squares elevation time series  $E\left(\frac{\partial^2 z_0}{\partial x^2}\right)$ ,  $E\left(\frac{\partial^3 z}{\partial x^2 \partial t}\right)$  compared to the original Smith et al. (2017) implementation by factors of 5 and 10, respectively. These radar altimetry measurements complement SAR observations of
  - integrated vertical displacement, which we use together to understand the character of new lake drainage activity.

**70 3 Results: new observations of lake activity**

A complete chronology of progressive thinning and lake activity across Thwaites Glacier from the extended CryoSat-2 time series is shown in the video supplement (Movie SVI). These new observations reveal that the upper Thwaites Lakes, Thw170 and Thw142, drained in 2017, filling Thw124 (Figs. 1 & 2). The SAR-derived elevation-change data show that the largest lake, Thw124, filled by 1.9km3 during the 2017 drainage, roughly balancing the volume that drained from Thw142 (0.6km3) and

- 75 Thw170 (1.4km3). The quarterly CryoSat-2 results show less clear evidence of water budget balance (Fig. 2, Fig. S2), which may be due to the degree of smoothing used in producing the time series. From CryoSat-2 elevation-change data, between 2015 and before the 2017 drainage event, the areas inside the Thw124, Thw142, and Thw170 lake outlines increase in elevation, which is strong evidence of filling (Fig. 3, Fig S2).
- 80 The extended elevation time series (Fig. 3, Fig S2) also reveals the fill-drain cycle of two new lake systems: one in the western shear margin of Haynes Glacier and another in the western tributary of Thwaites Glacier (Fig. 1). From these combined observations, the western Thwaites tributary lake (WT) drained by 1.1km3 in 2013 and the Haynes Glacier lake system (HG) drained by 0.2km3 in 2017 (Figs. S1 & S2). Complete fill-drain cycles of the Haynes Glacier lakes and the western Thwaites Glacier lake are not observed in the existing altimetry record and the Haynes Glacier lakes do not discernibly refill after draining in 2017 (Fig. S2). The western Thwaites tributary lake, however, fills significantly at a rate of 0.1km3/yr after draining

in 2014.]

**Commented [AOH7]:** Added explaination of the fitting procedure and citation for more in-depth description of the fit.

Commented [AOH8]: Units were written incorrectly.

Commented [AOH9]: This phrase was reworded for clarity. Commented [AOH10]: Underused was deleted as these data are heavily used for vertical deformation measurements.

Commented [AOH11]: Added movie reference

Commented [AOH12]: Added for clarity

Commented [AOH13]: Also added for clarity and consistency.

**Commented [AOH14]:** This sentence was changed to attend to reviewer suggestion that the sentence was originally unclear as to if the lake volume change was a filling or draining rate.

**4 Discussion**

Cascading lake drainages have been observed under many Antarctic ice-stream systems (Wingham et al., 2006; Fricker et al., 2007; Siegfried et al., 2014; 2016). The positions of all identified lakes beneath Thwaites Glacier, including the new lakes in

- 90 the Haynes Glacier shear margin and western tributary of Thwaites Glacier, appear to be controlled primarily by the bed and associated surface geometry (Smith et al., 2017). There are large topographic ridges at the bed with corresponding expressions at the surface that are oriented orthogonal to flow and likely act as hydraulic baffles trapping water and sediments (Holschuh et al., 2020), causing hundred-kPa-scale deviations in basal traction (Joughin et al., 2014; Fig S5a). The weak basal shear stress in these till-draped basins combine with large scale catchment topography to promote variations in ice thickness and
- 95 surface slope that form large hydropotential lows (Fig. S4, Smith et al., 2017; Holschuh et al., 2020). In these hydropotential lows, the lakes remain disconnected from their neighbours as they fill until cascading drainages driven by the upstream lakes interrupt the background fill rate in the cycle. Densely-sampled SAR vertical displacement rates from 2017 ( $V_z$  in Fig. 2c) demonstrate this process, capturing the Thw170 drainage that initiated a combined drainage with Thw142 into Thw124 (Fig. 2).
- 100 The controls on lake filling are less clear. From the altimetry observations of the Thw170 fill cycle, the average fill rate is ~0.16km3/yr(Fig. S4). [This agrees with the fill rate (~0.14km3/yr) we calculate by routing modelled basal meltwater production (Joughin et al., 2009) down the glaciostatic hydropotential gradient (Shreve, 1972) into Thw170, but requires inflow of all melt water produced upstream into the Thw170 lake basin (Fig. S4).] The glaciostatic hydropotential also routes water around Thw170 into downstream lakes Thw142 and Thw124, but the fill rates associated with these flow paths (~0.44km3/yr and ~0.27km3/yr,
- 105 respectively) are much larger than the fill rates derived from surface height change (Fig. S4). [These discrepant observations may reflect limitations of the static hydropotential assumption and the modelled catchment meltwater budget, but also suggest current bed-elevation models do not resolve small-scale (<1 km) bed topography important for routing subglacial water into the upper most lake, Thw170].

**4.1 Lake impact on ice flow and coupled drainage morphology**

- 110 The inland SAR and GNSS observations show a general pattern of acceleration at the LTHW and UTHW sites, consistent with an increase in driving stress due to inland propagation of thinning caused by ungrounding and loss of ice-shelf buttressing (Rignot et al., 2014; Joughin et al., 2014). Figure 3 shows the velocity anomaly at LTHW after subtracting the 2010 glacier velocity (340m/yr). At LTHW, this secular trend is punctuated by two signals associated with the Thwaites Lakes drainage events, first in September 2012 and again in May 2017 (Fig. 3). During the 2012 drainage documented by Smith et al. (2017),
- 115 surface velocities initially spike by 2% over a several-day period but then decline by 3% over the following 6 months. Loss of receiver power interrupted this record in March 2013. When the receiver began telemetering data again in 2015, the relative change in position suggests the speed anomaly in 2014 remained below what would be expected from the 2010–2012 trend. From January 2016 to May 2017, the LTHW receiver continued to accelerate at a rate of 4m/yr2. As Thw142 and Thw170 drained

**Commented [AOH15]:** Figure added that captures null change in velocity across the lake basin in the supplement.

**Commented [AOH16]:** Sentence changed to improve readability. Specifically, reworded to avoid juxtaposition of "changes" with static hydropotentical.

Commented [AOH17]:

in the austral winter 2017, filling Thw124, there was a nearly step-wise 1% increase in glacier speed at LTHW (Fig. 3). Speed
 remained elevated at the LTHW site after Thw124 stopped filling, coinciding with a 2-degree shift in ice-flow direction to the grid-north (clockwise), toward Thw124 (Fig. 3); however, this speed change is imperceptible in distributed velocity maps before and after Thw124 filled in 2017 (Fig. S3). The UTHW site also exhibits subtle velocity fluctuations with a magnitude less than 1% of the mean velocity (105m/yr) after correcting for the spatial ice-velocity gradient. These fluctuations are small relative to the background acceleration we observe at the UTHW site, 0.75m/yr2.

125

[The transient velocity anomalies that depart from the trend observed at LTHW are likely related to subglacial lake dynamics.] During lake filling in 2017, the areal extent of the Thw124 lake increases. This increase should reduce traction at the lake boundary, both at the margins (near the LTHW site) of the lake but also inside the lake as unresolved topographic pinning points are submerged. In 2017, changes in basal traction occur almost immediately, causing a step increase in velocity (Fig.

The positive GNSS acceleration observed in September 2012 (Fig. 3) suggests that Thw124 began to drain in 2012 before the quarterly-resolved subsidence associated with the lake drainage becomes distinguishable in the CryoSat-2 surface elevation

- 135 time series (Fig. 3). The cause of the acceleration we observe at Thw124 before the bulk drainage cannot be unambiguously attributed to a discrete set of processes with these data, but the finite duration of the velocity increase (~10 days) suggests that a distributed drainage system may have been established at the downstream edge of the lake preceding bulk drainage. Glaciostatic hydraulic water routing indicates that Thw124 would likely drain at the grid-south edge of the lake near the LTHW GNSS receiver (Fig. S3). Enhanced lubrication outside the low-drag Thw124 basin as the lake begins to drain likely increases
- 140 local slip and drives the subtle change in ice-flow direction that we observe in the austral winter of 2012 before the peak drainage in 2013, when flow direction shifts back to the mean flow direction between 2010-2012. Non-steady effective pressure likely also affects the basal shear stress as the drainage system initially forms, then empties and closes. Similar to lake drainages under the Siple Coast ice streams (Siegfried et al., 2014; 2016), changes in basal slip speed due to lake activity are small (~1-2%) relative to the average sliding speed.
- 145

Regardless of the exact drainage mechanism, the locations of the lakes are governed by ice-flow response to the underlying bed topography, which promotes hydropotential basins that form as ice flows over ridges. Once a connected drainage begins, differences in water pressure in the conduits between lakes promote efficient drainage down the hydraulic gradient. During drainage, each lake is likely in local equilibrium with the ends of the conduits that directly connect to it; however, the large

150 hydropotential differences between lakes (~1MPa) cannot equilibrate along the entire length of the drainage path, which implies that the pressure difference over the length of the conduit is likely more important in determining whether water flows into or out of the lakes than the small variations in the hydraulic potential in the lakes as they fill and drain. Conversely, Commented [AOH18]: Reworded for clarity.

**Commented [AOH20]:** This sentence was changed in response to reviewer 2

**Commented [AOH21]:** Reworded for clarity.

**Commented [AOH22]:** Revised to be consistent with use of subsidence used in methods section.

**Commented [AOH23]:** Reworded this sentence for clarity due to suggestion of reviewer 1.

**Commented [AOH24]:** The previous sentence was omitted in response to concern from coauthors over the statement's validity.

drainage of upstream lakes may disrupt the steady-state drainage morphology by temporarily increasing the hydraulic conductivity of the conduits, bringing additional water into the lower lakes and allowing drainage from adjacent lakes that

155 share the same downstream conduit. The processes governing changes in subglacial hydraulic connectivity are poorly understood, but likely depend on the local dynamic hydropotential and the evolution of conduit morphology, which we do not try to infer from our observations of the fill-drain cycle. We only note that these processes likely contribute to the variability in lake fill-drain levels observed elsewhere over multiple fill-drain cycles (Fig S2., Siegfried et al., 2014; 2016), which cannot be explained by the evolving glaciostatic hydropotential alone.

**160 4.2 Implications for basin-wide change**

[The velocity changes in Figure 3 that we attribute to lake drainage events represent the dominant, albeit small (<~3%), inland sub-annual velocity variability. The lakes sequester water and thus likely play some not well-understood role in maintaining distributed low effective pressures that control long-term flow rates, but the fill-drain cycles have little transient effect on the flow behaviour of the overlying ice. On decadal timescales important for understanding ice-sheet behaviour and contribution

- 165 to sea level, the lakes do not appear to control inland ice-flow variation. Static inversions for bed resistance before and after the 2017 lake drainage event (see Supplement Section 4 for details) are not sensitive to the subtle surface velocity changes we measure with GNSS (Fig. S5), and the lakes identified by Smith et al. (2017) have no discernible effect on ice velocity at the UTHW site. These new observations suggest that the observed speed-up at the grounding zone of the main trunk of Thwaites Glacier following the 2013 drainage (Smith et al., 2017) was associated with warming ocean conditions following anomalous
- 170 Amundsen Sea wide ocean cooling from 2012-2013 (Christianson et al., 2016). These warm ocean conditions likely enhanced sub-ice-shelf melt and led to increased ungrounding and acceleration. Our observations and model experiment (Supplement Section 4) invalidate proposed geoengineering solutions that seek to drain large volumes of water from beneath Amundsen Sea Embayment glaciers to increase basal resistance (Moore et al., 2018). These results further demonstrate that capturing the details of lake fill-drain cycles, and at least some elements of the associated basal hydrology system, may not be that important
- 175 for modelling Thwaites Glacier's contribution to sea level on decadal to centennial timescales.

**5 Conclusions**

We document the temporal change in velocity and elevation far from the grounding zone in response to the steepening of Thwaites Glacier and three distinct systems of active lakes: one on the main Thwaites Glacier trunk, another in the western shear margin of Haynes Glacier, and a third in the westernmost tributary of Thwaites Glacier. At the LTHW GNSS site, over

180 one hundred kilometres from the grounding line, ice velocity has accelerated at a nearly constant rate over the last decade. This background acceleration was interrupted in 2012 by the connected drainage of lakes Thw124, Thw142, and Thw170, and, in 2017, by the partial filling of Thw124 via drainage of Thw142 and Thw170. Our observations suggest that the transport of ~2 cubic kilometres of water beneath Thwaites Glacier, which represents approximately half the annual basal meltwater production for

**Commented [AOH25]: Reworded for clarity.**

 Commented [AOH26]: Sentence added that captures the sentence removed from thee paragraph above. This was an organizational decision, to move the sentence to hypotheses and and discussion of basin wide change.

Commented [AOH27]: Added reference to supplement.

**Commented [AOH28]: Added supplement section.**

[revised manuscript text omitted]